# Comment on Marasca et al. Teledermatology and Inflammatory Skin Conditions during COVID-19 Era: New Perspectives and Applications. *J. Clin. Med.* 2022, *11*, 1511

**DOI:** 10.3390/jcm11144063

**Published:** 2022-07-14

**Authors:** Francesco Borgia, Federica Li Pomi, Clara Alessandrello, Sebastiano Gangemi

**Affiliations:** 1Department of Clinical and Experimental Medicine, Section of Dermatology, University of Messina, 98125 Messina, Italy; federicalipomi@hotmail.it; 2School and Operative Unit of Allergy and Clinical Immunology, Department of Clinical and Experimental Medicine, University of Messina, 98125 Messina, Italy; clara.alessandrello@outlook.it (C.A.); sebastiano.gangemi@unime.it (S.G.)

We welcome an enthusiastic article titled “Teledermatology and Inflammatory Skin Conditions during COVID-19 Era: New Perspectives and Applications” by Marasca et al. [1].

The restrictions and social distancing imposed by the actual pandemic determined the reduction of dermatological consultations by approximately 80–90% [2] and the closure of non-urgent outpatient clinics devoted to caring for chronic, severe, inflammatory skin diseases [3]. The management of these diseases is challenging for dermatologists since their relapsing chronic clinical course is associated with a great impact on quality of life and these patients require periodic follow-up and personalized therapies [4]. Recently it has been pointed out that patients with chronic dermatological diseases, such as psoriasis, have greater adherence to therapy with biological drugs, if followed up via teleconsultation. This strategy could guarantee excellent results if applied to other chronic dermatological diseases such as atopic dermatitis and hidradenitis suppurativa in terms of time and money savings and convenience [5]. In this context, teledermatology has represented a prompt solution acting as a substitute for face-to-face visits for many skin diseases, whereas, in a normal setting this virtual system could be used as a supplement to standard dermatological visits [6]. This is particularly relevant to dermatology as the visual examination is the keystone in identifying most dermatological pathologies [7], also in the perspective of the continuation of the current pandemic and the growing possibility of future pandemics, such as the new Monkeypox cases emerged in May 2022. The advantages of telemedicine concerning the shortening of geographical distances and time limits, as well as the lack of medical personnel often present in the most disadvantaged areas, allows people to obtain health care without leaving home. The implementation of telemedicine services could guarantee better assistance to patients who live geographically distant from the Hub centers (centers of reference that concentrate on highly complex assistance). Furthermore, telemedicine would facilitate the integrated management of the patient between the Hub centers and the Spoke centers (a territorial network of outpatient services that are responsible for selecting and managing patients and sending them to the reference centers when a certain clinical-assistance severity threshold is exceeded). Another useful tool to follow the patient remotely could be the use of apps on smartphones or PCs, making the use of teledermatology easier and more immediate. A useful app should require the inclusion of photographs to be able to calculate the severity index scores, the quality of life, to evaluate the quality of sleep (sleep loss), and the intensity of symptoms. Moreover, it should also be possible to make e-lab prescriptions and e-drug prescriptions. In this way, it would be possible to assess and stage the patient at a distance and evaluate the most appropriate therapy. Moreover, the app may also send alerts to remind the patient to periodically upload photos, score results, and periodically prescribed follow-up blood test results. Finally, one last aspect to consider is that the use of telemedicine on a large scale would promote the reduction of health costs and waiting lists for dermatological visits in specialized centers. Teledermatology also sets limits. First, the patient must be equipped with devices with a good camera, they must also be smart, and capable of using technological devices. Within an elderly population, this aspect can represent a limit difficult to overcome. In addition, patients should have good internet access and the use of online platforms could be invalidated by connection problems or overload. Another aspect to consider is the physician-patient relationship, which could appear more detached from the patient. In this way, patients, especially the more fragile ones, could be wary of using telehealth and abandon follow-ups in search of in-person visits. For this reason, it would be advisable to associate a periodic assessment of the quality of remote assistance and periodically carry out visits in the presence to reassure the patient. COVID-19 has certainly been a tragedy that has struck the world and is still affecting it, but it has certainly given a driving force to the spread of the practice of telemedicine, which we hope will be used more and more for selected classes of patients. Furthermore, COVID-19 must be a lesson for the possible development of other future pandemics. The remodeling that medicine has rapidly undergone must not be forgotten but used as taught for the advent of possible other pandemics. Finally, telehealth reduces barriers to accessing healthcare for people with disabilities, the elderly population and caregivers needing to find childcare for other children, allowing patients to have a specialist evaluation comfortably in their own homes [8]. We agree with the analysis carried out by Marasca et al. Telemedicine is a precious tool that will find application in future clinical practice, fully integrating with face-to-face medicine. To achieve a wider use of telemedicine, it is necessary to improve the digitization process already underway in healthcare environments, which the advent of the COVID-19 pandemic has already sped up.

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
