# Peer review of "Comment on Marasca et al. Teledermatology and Inflammatory Skin Conditions during COVID-19 Era: New Perspectives and Applications. J. Clin. Med. 2022, 11, 1511"

_jcm, 2022, doi:10.3390/jcm11144063_

Round 1

Reviewer 1 Report

This is an interesting comment on the implementation of teledermatology in COVID-19 Era and its potential application/usefulness in the post-COVID19 period.  The manuscript is well-written  and may be published as it is.

Author Response

Dear Reviewer, 
thank you for your appreciation for our paper. 

Best regards. 

Reviewer 2 Report

The authors reported a commentary on the recent manuscript entitled “Teledermatology and Inflammatory Skin Conditions during COVID-19 Era: New Perspectives and Applications” underlying the advantages of telemedicine and future perspectives. The manuscript is interesting and well written. I have only few suggestions. My comments:   -  Teledermatology can also be a useful tool also for its cost effectiveness and for reducing waiting lists. Moreover, it can be a valuable strategy for patients undergoing biologic treatments on manteinance therapy. Please discuss - Teledermatology may be a useful tool also after Covid-19 era. You should read and cite "Megna M, et al. Teledermatology: A useful tool also after COVID-19 era?. J Cosmet Dermatol. 2022;10.1111/jocd.14938. doi:10.1111/jocd.14938"

Author Response

Dear Reviewer,

Thank you kindly for your observations. As suggested, we discussed teledermatology as a useful tool for its cost-effectiveness, for reducing waiting lists and as a possible strategy for patients treated with biological treatments. We also read with great interest and cited "Megna M, et al. Teledermatology: A useful tool also after COVID-19 era?. J Cosmet Dermatol. 2022;10.1111/jocd.14938. doi:10.1111/jocd.14938"

Hoping these match your remarks, we wait for any further suggestions.

Best Regards